# Sheltered Employment Centres: Sustainability and Social Value

**María Jesús Segovia-Vargas** [1] , **María del Mar Camacho-Miñano** [2] , **Fernanda Cristina Pedrosa Alberto** [3]
**and Vera Gelashvili** [4],*

1   Department of Financial and Actuarial Economics & Statistics, Faculty of Business Administration and
    Economics, Campus of Somosaguas, Universidad Complutense de Madrid, Pozuelo-de-Alarcón,
    28230 Madrid, Spain; mjsegovia@ccee.ucm.es
2   Department of Accounting and Finance, Faculty of Business Administration and Economics,
    Campus of Somosaguas, Universidad Complutense de Madrid, Pozuelo-de-Alarcón, 28230 Madrid, Spain;
    macamacho@ccee.ucm.es
3   Department of Accounting, Coimbra Business School ISCAC, 3045-601 Coimbra, Portugal; falberto@iscac.pt
4   Department of Business, Faculty of Law and Social Sciences, King Juan Carlos University,
    28032 Madrid, Spain
*   Correspondence: vera.gelashvili@urjc.es

**Abstract:** Sheltered employment centres are social enterprises where at least 70% of their workers have disabilities. They are a way of helping people with disabilities to work in good working conditions and of allowing disadvantaged people to live a full life. However, some people criticise these businesses for being ghettos where public subsidies are used inefficiently. Our paper aims to test if this criticism is valid by analysing whether these companies provide social and economic value to society in return for public funding and are also economically sustainable over time. Using a sample of 997 Spanish sheltered employment centres, a descriptive analysis of the main variables has been carried out. Additionally, the results of a PART algorithm show the relationship between these companies and economic sustainability. Our findings corroborate that these firms are economically sustainable and, at the same time, socially sustainable. These results highlight the great work that such companies perform for society and the country's economy.

**Keywords:** sheltered employment centres; sustainability; economic sustainability; social sustainability

## 1. Introduction

The purpose behind this paper is to analyse the social value created by sheltered employment centres from being sustainable from a social and economic perspective. Our study focuses on sheltered employment centres in Spain, defined as businesses in which at least 70% of their workers have disabilities (with an official certification of a degree of disability of more than 33%). Once a company has been founded, it needs to apply to the Regional Government for recognition as a sheltered employment centre, which it will receive if the legal requirements are met. If necessary, the administration studies the business project and determines whether it should be recognised, registering it in the regional register for this type of centre to allow it to receive public funding. The government pays social security contributions for workers, a portion of their salaries and some additional grants to support this kind of social firm.

In Spain, the concept of a sheltered employment centre was first introduced in 1982 to increase employment among people with disabilities. The first step in this process was the Law for the Social Integration of People with Disabilities (LISMI) [1]. As they are recognised as socially responsible companies, they receive public funding to help their foundation, business payments, business and worker social security contributions, maintenance of jobs, etc. (Royal Decree 1/2013, 29th November).

However, some critics are against this business model [2,3]. First, some people think of these companies as ghettos where people with disabilities are kept apart from other

workers in abnormal and protected working conditions, without any natural inclusion in society. As sheltered employment centres receive significant public funding from the government and other institutions such as the European Union, they have advantages and can hire staff at a lower cost than other firms [4]. Second, some critics believe that having a subsidised employment incubator is the goal itself, rather than a step towards full inclusion in the normal labour market [2].

Therefore, we aim to test if these companies contribute a social value that offsets the public funding they receive when hiring people with disabilities. If this social value is positive, we would have found an argument to counter their biggest critics. Using a sample of 997 sheltered employment centres in Spain, our results confirm that these companies are an example of social sustainability. At the same time, they generate economic sustainability through their activities. These results have been obtained from a descriptive analysis of these companies' economic and financial data. A PART algorithm was then used to measure the contribution of sheltered employment centres to economic sustainability. Our results highlight the importance of these companies for people with disabilities, society and the country's economy.

This paper is organised into the following sections: Section 2 reviews the existing literature and sets out the hypothesis. Section 3 explains the research design, including the sample and methodology, while Section 4 describes the descriptive statistics and discussion results. Section 5 summarises our findings and sets out the main conclusion and social and economic implications.

## 2. Literature Background

### 2.1. Sustainability and Its Importance

The United Nations General Assembly first defined sustainability in the report by the World Commission on Environment and Development. According to this report (also known as the "Brundtland Report" or "Our Common Future Report") which was written by Brundtland et al. [5], "sustainability is the development that meets the needs of the present without compromising the ability of future generations to meet their own needs". Starting from this definition and after the first appearance of the concept of sustainability, several authors have tried to add to or interpret this idea more broadly in numerous disciplines and various contexts [6–8]. There is agreement about the lack of a consistent definition on this topic in the literature. Despite this, the concept of sustainability has been, and still is, fashionable in political agendas and the organisational strategies of big companies [9], above all today with the worldwide movement against climate change. Therefore, governments and national and international organisations are increasingly promoting sustainable development as an essential element to improve a country or individual's environmental, social or economic situation.

Generally speaking, sustainable development is divided into three dimensions: economic sustainability, social sustainability and environmental sustainability [10–12]. In contrast to the overall concept of sustainability, these three dimensions of sustainability are well analysed and defined in studies [12,13]. The dimensions of sustainability are defined as follows.

Economic sustainability refers to the process of allocating and preserving scarce resources while ensuring positive social and environmental outcomes [14] and responsibly generating profitability over the long term. To achieve economic sustainability, companies must consider the implications of sustainable management, both internal and external. Therefore, they must consider the financial performance of the company, management of intangible assets, the company's influence on the broader economy and the control of social and environmental impacts [15].

Social sustainability is defined as "development that is compatible with the harmonious evolution of civil society, fostering an environment conducive to the compatible cohabitation of culturally and socially diverse groups while at the same time encouraging social integration, with improvements in the quality of life for all segments of the popula-

tion" [16]. According to Eizenberg and Jabareen [17], the concept of social sustainability is related to equity, safety, eco-prosumption (referring to the responsibility of society to reduce future risk) and sustainable urban forms. This means that the labour and social inclusion of vulnerable groups are considered social sustainability.

Environmental sustainability is one of the dimensions of sustainability that has been most discussed and analysed by researchers. It refers to the "maintenance of natural capital that includes renewable and non-renewable resources on the source side, and pollution and waste assimilation on the sink side" [18]. In other words, it means conserving and protecting the environment indefinitely. Nowadays, environmental sustainability programs are guided by actions to reduce the use of physical resources and involve the use of renewable resources rather than depletable resources [19].

Considering the definitions of each of the dimensions of sustainability, we can say that they are essential aspects in achieving a better life for everyone, the final aim of any strategy related to people with disability.

### 2.2. Social Value

The concept of social value is not new, although it has not been studied in the academic literature for a long time. The first definition of the concept was seen at the beginning of the 20th century when Schumpeter [20] analysed price theory and social value. Based on this study, "it is society and not the individual which sets a value on things, where exchange value is social value-in-use". This definition was in relation to the company and its price behaviour. The concept of social value has long been discussed subsequently, but there is no standard definition [21–24]. According to Wood and Leighton [23], social value refers to "wider non-financial impacts of programmes, organisations and interventions, including the wellbeing of individuals and communities, social capital and the environment". According to this definition, social value outcomes are challenging to measure and quantify. For this reason, they are typically described as 'soft' outcomes.

Companies with a social purpose mainly create social value [25–27]. Therefore, to measure the social value created by companies, it is first necessary to identify their social function [28]: that is, the reason why social enterprises pursue activities such as providing employment for people with disabilities or other disadvantaged persons, promoting and providing education for underprivileged children or performing activity that helps tackle social problems [29]. Therefore, social enterprises are run by people whose main objectives are to improve society [30]. In addition, the importance of social enterprises to the economy must be underlined. A study produced by the European Union (2021) (Available online: https://www.interregeurope.eu/fileadmin/user_upload/plp_uploads/policy_briefs/The_social_economy_and_support_to_social_enterprises_in_the_European_Union_Policy_brief.pdf, accessed on 15 April 2021) has shown that almost 14 million jobs were created in social firms in European countries and that the social economy accounts for 8% of European Union GDP.

When analysing social enterprises, the concept of "blended value" must be considered. This concept refers to the conceptual framework where different organisations, among them social enterprises, are evaluated based on their economic, social and environmental impact (Available online: https://www.blendedvalue.org/framework, accessed on 15 April 2021). According to Emerson [31], the person who invented and developed the concept of blended value, social value can be combined with quantitative financial or economic value. The generation of economic or monetary value for some positively affects the creation of social value for others. An empirical study by Mazhar and Siddiqui [30] on blended value (this study only looked at social and financial value creation) showed that the number of employees is a significant variable when analysing business success in relation to financial performance. However, this variable has still not been associated with social success.

There is no doubt that social enterprises do an excellent job in promoting and raising the profile of social issues, providing social services and tackling social exclusion [32]. It is, therefore, increasingly necessary to measure the social value generated by them.

In the literature, some papers summarise the principal methodologies for analysing social value that have been used worldwide [22,33,34]; among these are:

- Cost–Benefit Analysis (CBA) estimates the weaknesses and strengths of projects that create social and economic value. This method is based on the valuation of investment projects, where a monetary value is placed on the expected benefits from the project. This is then compared with the costs expected to be incurred in the future. Benefits are understood as creating employment, a positive effect on the local economy, improving the population's health or quality of life, revenues, indirect savings, etc. In the meantime, costs could be staff wages, training, rent, purchase of equipment, publicity and promotion, etc.

- Social Accounting refers to the methodology through which all of a company's quantitative and qualitative information is analysed. Under this methodology, the whole company is studied rather than just individual projects, as is the case with CBA. Social accounting is calculated at the end of a financial year, and the company collects all the information, audits it and presents it in a social report.

- Social Return on Investment (SROI) is the best-known methodology for analysing the social value created by a company, but it must be considered that it is impossible to calculate it with only the data provided in annual reports. The SROI methodology includes CBA and Social Accounting measurement elements. Therefore, it is a more sophisticated and complex approach. A distinction can be made between two different types of SROI: (a) Evaluative—conducted in a retrospective manner and based on actual results, and (b) Forecast—based on predictions of how much social value will be created in the future if the estimated effects are achieved.

- Basic Efficiency Resource (BER) is a cost-effective approach that aims to provide a more straightforward framework for evaluating complex programmes, campaigns or activities. It is based on using SROI to evaluate specific elements where there is an impact, considering the resources and the performance of the analysed units compared to others. This methodology uses the inputs and outputs obtained through interviews and surveys with the company's external and internal users.

Although these different models measure the social value created by companies, it is not easy to obtain the data to calculate them. Moreover, companies themselves have problems with valuing their social contribution, mainly due to the prominence of economic and financial indicators which only have an instrumental value in this type of company [35]. Consequently, three alternative perspectives are involved at the monetary level of social value: the social value generated through economic activity; the socio-economic return for the public body; and the specific social value generated for specific groups of stakeholders [36].

Given that sheltered employment centres regularly receive subsidies from public bodies, this paper aims to estimate the economic return via taxes paid to the various levels of government. Financial support for people with disabilities, and therefore for sheltered employment centres, must be framed within the principle of equity; our society wants to have a welfare system that allows people with disabilities to enjoy similar opportunities to those of the rest of the population.

The cost–benefit analysis methodology is applied to calculate this return, subtracting the outcomes generated for the public purse from the costs that the public bodies may have incurred in funding the centre [37,38].

This return indicator, linked to the contribution made by these companies through the hiring of employees with disabilities, shows that the government receives an economic return from the hiring of these groups. However, it is also essential to highlight the intangible value for the government and these people that comes from them joining the labour market and their social inclusion through employment (a very complex issue to explain with an economic indicator). Therefore, employment for disabled people has a direct benefit on social cohesion and other indirect indicators such as the family spending on items such as food and leisure, to give two examples [39]. In other words, this return

shows that governments receive an economic return from hiring these groups and save a consequent social expenditure.

According to the literature on social value, this is created by companies or individuals that improve the situation of individuals or a specific group of people, respect and help maintain the environment or generate prosperity or economic wealth for the company and society. Depending on the situation, the community's needs and individual changes, social values can be adjusted or reorganised over time. Thus, companies or individuals involved in social value creation should continuously learn how to improve and create social value of true benefit to society.

### 2.3. Sheltered Employment Centres

Around Europe, various programmes support and promote the employment of people with disabilities. One of the most recent actions is the strategy on work, social affairs and inclusion facilitated by the European Union (Union of equality: Strategy for the rights of persons with disabilities 2021–2030. Available online: https://ec.europa.eu/social/main.jsp?catId=1484, accessed on 16 April 2021). This strategy aims to promote the rights of people with disabilities at the European sustainably level. Another way of combating labour discrimination against people with disabilities is through social firms [40]. Social enterprises put social objectives before economic ones, and if the company is profitable, those profits are often reinvested to promote the company's key social objectives [41]. Taking that into account, we can say that the objective of social enterprises is to exhibit a greater sensitivity and promotion in the social and labour insertion of groups at risk of social exclusion compared to what ordinary companies should have [42]. In addition, social enterprises have a high impact at an international level, since they contribute to the social and labour insertion of the most needy people and prevent discrimination on the basis of gender [43]. The importance of social enterprises in local and regional development is also worth noting as they can solve problems related to collective actions by creating networks and applying social norms to the situation [44]. Although they are important companies, there are several barriers for these companies. The study elaborated by [45] identified the following barriers for growth of social firms: value-based barriers (ethical value differences, growth philosophy and ethical principles), business models barriers (access to finance, access to human capital and identity authenticity) and institutional barriers (consumer culture and business norms).

Within the category of social enterprises, we have sheltered employment centres [46], which in some European countries are companies that promote the labour and social inclusion of people with disabilities. However, these companies are not well known, which means that their great work is not consistently recognised.

In Spain, the social companies that focus on employing people with disabilities are sheltered employment centres, occupational centres and the National Organization of the Blind [47]. The legal framework for sheltered employment centres has its origin in Law 13/1982, 7th April, on the Social Integration of People with Disabilities. In Royal Legislative Decree 1/2013 29th November, these centres were given their current definition as "those whose main objective is to perform productive work, regularly participating in the operations of the market, having the purpose to secure paid employment and the provision of the personal and social adaptation required by their workers with disabilities, at the same time as being a means to include the greatest number of people with disabilities in the ordinary labour market". The Law stipulates that at least 70% of the staff of these firms must consist of workers with disabilities. Barea and Monzón [48] state that sheltered employment centres currently account for the highest proportion of labour inclusion of people with disabilities in Spain. Although the inclusion of people with disabilities in ordinary companies would be the ideal solution, a disability can hamper their full inclusion in this type of company, especially if their degree of disability makes them very dependent. Therefore, through these companies, people with a high degree of disability can acquire skills and experience. If they cannot find a job in the ordinary labour market, they can turn

to one of these centres to find their first job and gain skills and experience [49]. Moreover, the support, training and assistance from these companies can facilitate their full labour and social inclusion [50,51].

Several studies have followed the evolution in Spain of this specific type of business over the last two decades. All of them have concluded that the number of such businesses and the number of people with disabilities who have found work through them, have increased markedly [3,52–57]. According to Penabad et al. [51] the population with disabilities in Spain forms a large group with low labour-force participation, with activity and employment rates far below those of the non-disabled population. Therefore, the existence of sheltered employment centres helps the labour inclusion of people with a disability, which significantly reduces their risk of unemployment. In addition, they also provide protected employment; therefore, although they operate in the open market, they respect their social-support purpose and continuously encourage personal and social adjustments for their workers [49].

Sheltered employment centres receive public funding in a different way, such as one-off grants for hiring people with disabilities, eliminating architectural barriers, adapting workplaces, covering social security contributions for disabled workers, etc. However, like other ordinary companies in the market, they must ensure their economic and financial viability [57]. This means that they need to be profitable and competitive in national or international markets, and as such, these firms make a direct contribution to the social economy of the country.

Bearing all these things in mind, we can postulate that these companies directly contribute to social and economic sustainability. In terms of social sustainability, sheltered employment centres make employment and support available to people with more significant difficulties accessing the labour market. They also promote the social and labour inclusion of people with disabilities. Since social sustainability aims to help social sectors or populations that are more disadvantaged than others or that are unprotected in some way, these companies promote and implement social sustainability daily. In the case of economic sustainability, we can argue that the contribution of these companies is also quite significant since the contribution of these companies is quite substantial. Empirical studies have shown that these companies generate profits (although not very high profits) and can survive in the market, indicating that they efficiently manage their resources and sustainably develop profitability over the long term [57]. Other theoretical studies [49,51] have argued that sheltered employment centres incorporate economic and social aspects when assessing performance, which shows their clear participation in social and economic sustainability.

In addition to this, sheltered employment centres create social value by creating abilities and empowering people to improve their social and working conditions.

### 2.4. Research Questions

Over the decades, social enterprises have demonstrated the great work they do to improve various aspects of society. There are different types of social enterprise with distinct objectives. However, all of them are focussed on creating a better world for the most disadvantaged people and for society in general. In Spain, there are three different types of social enterprise: sheltered employment centres, social cooperatives and insertion companies [46,58].

The importance and evolution of sheltered employment centres has been studied over the years [52,54–56]. Authors have contributed to defining the value created by these companies [49,57], but to the best of our knowledge, there are no studies that examine sheltered employment centres in relation to sustainable development (specifically social and economic sustainability) and social value creation. Moreover, lessons about sustainability have also been published [8,11,12]. In recent years, there have also been various initiatives promoted and proposed by governments, international organisations and researchers [59,60]. All these reports and studies have found that sustainability is a crucial

factor in improving social progress and stable economic growth, promoting equality and environmental protection and enhancing sustainable development at all levels of society.

In addition, an increasing number of attempts are being made to measure the value created by social enterprises [26,30]. However, it is currently a challenge for companies and analysts to measure the social work done by these enterprises [22,35]. According to Mulgan [22], there are different ways to measure the social value created by companies. However, it should be noted that this is not easy to calculate. It requires extra information, often not found in the financial statements or other reports that companies produce each year.

With that in mind, in this study, we want to analyse whether social enterprises, such as sheltered employment centres, focussed on the social and labour market inclusion of people with disabilities, promote or contribute to sustainable development. Additionally, we want to check if it is possible to measure the social value created by these companies using the information in their annual accounts.

Therefore, the following research questions are analysed:

RQ1: Do sheltered employment centres contribute to the creation or improvement of social and economic sustainability?

RQ2: Do sheltered employment centres create social value through their operations and activity measured through the return to the public administrations?

To answer these research questions, the first step was to complete a literature review on the objectives of these companies. The next step was to analyse their annual accounts to see if they contribute to economic sustainability based on economic and financial indicators. The last step was to test whether a calculation of the social value created by these companies would be possible.

## 3. Data Collection, Variables and Methodology

The sample for this study is all the sheltered employment centres in Spain found in the official register held by SEPE (Spanish Public Employment Service). There are more than 2000 sheltered employment centres in our country, although there is no combined register since they operate in different autonomous regions. We visited SEPE in person, and they gave us a list of all Spanish sheltered employment centres. We did an Internet search using their names to find their tax code (CIF in Spanish). With the tax code, we searched for our initial list in the SABI database. A total of 997 firms were found there. We extracted their key figures and their financial statements from 2015 to 2019. Through these, we obtained the basic data on each company and detailed economic and financial reports.

The variables used were those that allow us to analyse the current position of sheltered employment centres. The list of all the variables used in the study is shown in Table 1.

The selection of variables was carried out considering their importance for the economic and social sustainability of these enterprises. For the study, the dependent variable was return on assets (ROA), which measures the profitability of a company in the financial year [61].

The variables age (number of years before May 2021 that the company was founded) and size (the number of employees) were studied to analyse their relationship with the profitability of the company. Some studies have found a positive relationship between ROA and the two variables [62,63]. Bearing in mind the main characteristics of sheltered employment centres, the business sector variable was divided into two parts: service and manufacturing companies, since previous studies have shown that most of these centres operate in the service sector. We chose the variable sales growth because research carried out by Miethlich and Oldenburg [64] has shown that social inclusion, as part of the Corporate Social Responsibility strategy, contributes in a positive way to sales growth. In addition to these ratios, the fundamental economic and financial ratios necessary for the company's profitability were analysed [62].

**Table 1.** List of variables used in the study.

| ID | Variables | Definition |
|---|---|---|
| AGE | Age | The number of years since its foundation |
| SIZE | Size | The number of employees |
| SECT | Sector of activity | Manufacturing companies 0; Service companies 1 |
| CURR-R | Current Ratio | Current Assets/Current Liabilities |
| INDB | Indebtedness | Total Liabilities/Total Equity |
| SOL | Solvency Ratio | Total Liabilities/Total Assets |
| ROA | Return on Assets | Net Result/Total Assets |
| ROE | Return on Equity | Net Result/Total Equity |
| SALES-G | Sales Growth | Sales t–Sales t-1/Sales t-1 |
| SALES-E | Sales per employee | Sales/Number of employees |
| F-RISK | Financial Risk | Financial Expenses/Sales |
| CF | Cash Flow | Profit tax—Capital subsidies |
| SUBS | Public subsidies | Amount of money received from public institutions |

Source: own elaboration.

The first step in the data analysis was a descriptive analysis of the companies and the variables studied. The descriptive analysis was carried out for the last five years, to give a better picture of these companies. After that, a PART algorithm was run to identify the relationship between sheltered employment centres and economic sustainability. These two methods of analysis have provided a clear picture of these companies and their sustainability.

## 4. Results and Discussion

### 4.1. Descriptives

Firstly, we can look at the size of the companies analysed. The criteria established by the European Union to classify companies based on size were used (Commission Recommendation of 6 May 2003. Available online: https://eur-lex.europa.eu/legal-content/EN/TXT/?uri=CELEX:32003H0361, accessed on 16 April 2021). Companies with fewer than ten employees and an annual turnover or annual balance sheet total not exceeding EUR 2 million are considered micro-enterprises. A small company is one with 10–49 employees and an annual turnover or annual balance sheet total not exceeding EUR 10 million. Medium-sized companies have 50–249 employees and an annual turnover not exceeding EUR 50 million or an annual balance sheet total not exceeding EUR 43 million. Finally, large companies have more than 250 employees and an annual turnover exceeding EUR 50 million or an annual balance sheet total exceeding EUR 43 million.

As shown in Figure 1, most of the sheltered employment centres operating in Spain are micro-companies. Small enterprises represented 30% of the total sample, accounting for 300 sheltered employment centres. In total, 217 companies analysed were medium-sized companies, and lastly, there were 59 large companies, around 6% of our sample.

The number of employees ranges from micro-companies with only one worker up to one company with more than 7000 employees. On average, they have 74 employees, although we must remember that the standard deviation for this variable is relatively high. Another vital piece of data when analysing sheltered employment centres is their age. On average, these companies have been operating for almost 19 years, which means that they have experience in the market. The descriptive results for the age and number of employees variables are shown in Table 2.

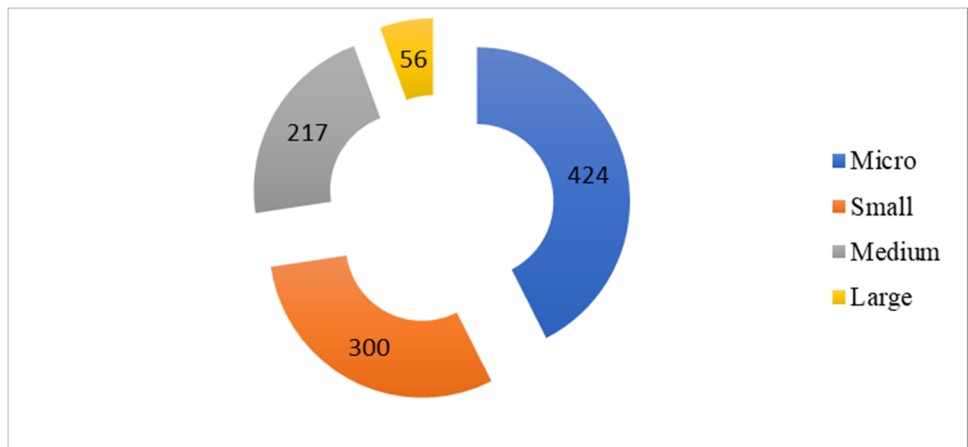

**Figure 1.** Size of sheltered employment centres. Source: own elaboration.

**Table 2.** Sheltered employment centres by age and number of employees.

|                           | Min. | Max.  | Mean  | St. Dev. |
|---------------------------|------|-------|-------|----------|
| Age (30 May 2021)         | 7.70 | 63.95 | 18.85 | 7.52     |
| Nº employees (2019)       | 1    | 7140  | 74.03 | 299.41   |

Source: own elaboration.

Secondly, looking at the legal structure of sheltered employment centres (see Figure 2), the results show that 88% of the sample are limited partnerships. The main difference between a limited company and other types of companies is the fact that the liability of its members is limited to the capital contributed, i.e., the owners are not obliged to pay the company's debts out of their own capital.

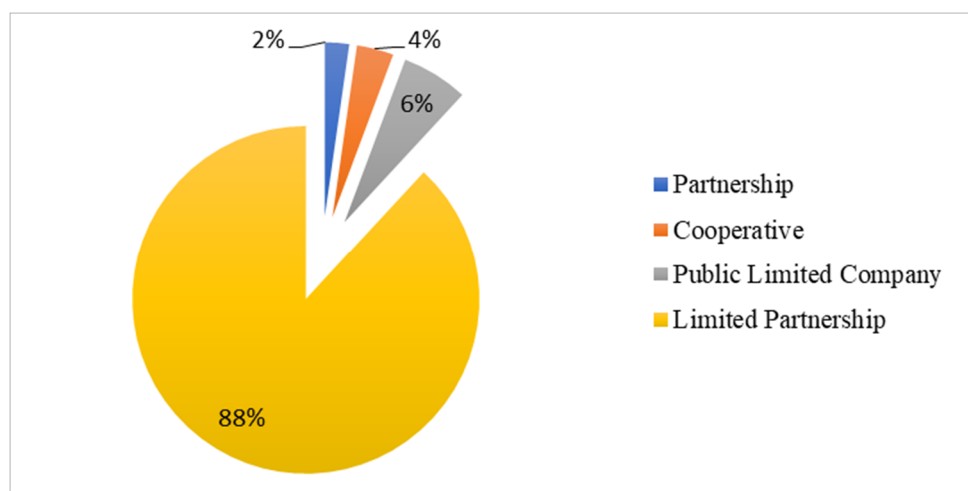

**Figure 2.** Size of sheltered employment centres. Source: own elaboration.

Next, the location and sector were analysed. We use the sector categories from the NACE (Statistical Classification of Economic Activities in the European Community) (see Table 3). In Spain, there are 17 autonomous regions, and there is a different regulation about grants and subsidies for the sheltered employment centres for each autonomous region. Therefore, the variable location is important because it can condition the subsidies received by the sheltered employment centres.

**Table 3.** Sheltered employment centres by autonomous community and sector of activity.

| Autonomous Community (Location) | Nº | Sector of Activity (Industry) | Nº |
|---|---|---|---|
| Andalusia | 208 | Agriculture, livestock, forestry and fishing | 19 |
| Aragon | 59 | Industry (food, textile, manufacturing) | 176 |
| Asturias | 72 | Construction | 11 |
| Balearic Islands | 17 | Wholesale and retail trade; transport, . . . | 170 |
| Valencian Community | 92 | Hotel and catering | 25 |
| Canarias | 37 | Information and communication services | 89 |
| Castilla la Mancha | 57 | Administrative activities and auxiliary services | 306 |
| Catalonia | 99 | Health and social work activities | 103 |
| Community of Madrid | 157 | Other different activities | 98 |
| Extremadura | 77 | | |
| Galicia | 63 | | |
| La Rioja | 17 | | |
| Basque country | 23 | | |
| Other AC (<10) | 19 | | |
| Total | 997 | Total | 997 |

Source: own elaboration.

Most of the sheltered employment centres are in Andalusia, Madrid, Valencia and Catalonia. However, these do not account for all sheltered employment centres in Spain. As mentioned previously, there is no single public list of all the centres. In terms of the sector of activity of these companies, most of them operate in the service sector (306 companies provide administrative activities and auxiliary services), around 30% of the total sample. This could be due to many medium and large companies outsourcing this kind of service.

Additionally, these businesses are involved in accessible and very standardised activities that can be performed with less skill. The wholesale and retail trade, vehicle repair, transport and storage (170), together with different industry activities (food, textile and manufacturing) (176) are the activities most performed by these companies. Lastly, 103 centres are involved in health and social work activities.

The descriptive analysis of the ratios is based on the last five years so that we can see the evolution during this period. As shown in Table 4, the trend for the ratios significantly improves during the period analysed, by 50% comparing the positions in 2015 and 2019. The indebtedness and solvency ratios are relatively stable, particularly the second of these. The profitability analysis, completed using ROA and return on equity (ROE) ratios, shows positive values over the period. ROA in 2019 is almost double the figure in 2015. Although sales growth is significantly lower in 2019 than in 2018, it is crucial to highlight that sales per employee did not fall by the same proportion, which shows an improvement in the efficiency of human resources. This is in line with previous studies [64]. The financial risk is relatively low in the period, mainly in the later years. The cash flow resulting from the relationship with the government (this being the difference between corporation tax paid and capital subsidies received) shows that the change from a negative relationship in 2015 to a very positive one in 2019 is favourable for the government. This corroborates the idea that these businesses are sustainable from a public economic policy perspective and of benefit to society.

**Table 4.** Descriptive analysis of ratios.

| | Mean per Years | | | | |
|---|---|---|---|---|---|
| **Ratios** | **2019** | **2018** | **2017** | **2016** | **2015** |
| CURR-R | 4.35 | 3.41 | 3.00 | 2.95 | 2.87 |
| INDB | 1.10 | 0.70 | 0.51 | 2.13 | 3.19 |
| SOL | 0.73 | 0.69 | 0.75 | 0.73 | 0.69 |
| ROA | 2.65% | 1.83% | 3.64% | 2.67% | 1.41% |
| ROE | 15% | 27% | 30% | 12% | 24% |
| SALES-G | 0.18 | 0.52 | 0.12 | 0.13 | - |
| SALES-E | 37.37 | 49.40 | 50.80 | 107.06 | 96.38 |
| F-RISK | 0.15 | 0.03 | 0.10 | 0.27 | 0.22 |
| CF | 55.18 | 277.23 | 3.21 | 46.65 | −27.95 |

Source: own elaboration.

Table 5 shows the evolution of the source of income for the sheltered employment centres in our sample. The trend for private funding increased from 2015 to 2019. At the same time, public funding fell significantly, by over 60%. In addition, other operating income increased during the period analysed, with the same trend in the sales of products and services. Of the EUR 8446.99 thousand of total annual incomes, 92.9% comes from sales of products and services; 0.39% from public administration subsidies; and only 1.72% comes from private subsidies; and 17.19% from other operating income in 2019. Results in previous years are in line with 2019 results.

**Table 5.** Sheltered employment centres' source of income (in thousand).

| Source of Income | 2019 | 2018 | 2017 | 2016 | 2015 |
|---|---|---|---|---|---|
| Private Subsidies | 145.18 | 119.67 | 116.34 | 110.05 | 89.09 |
| Public Subsidies | 33.15 | 52.56 | 61.42 | 78.58 | 84.51 |
| Other operating incomes | 418.42 | 358.61 | 349.85 | 349.85 | 276.50 |
| Sales (products and services) | 7850.24 | 7678.54 | 7430.17 | 7345.42 | 7437.32 |
| Total annual incomes | 8446.99 | 8209.39 | 7957.78 | 7883.89 | 7887.42 |

Source: own elaboration.

As shown in Table 5, sheltered employment centres contributed more to the public finances than they received. Therefore, the balance is positive for the government. Consequently, encouraging this type of social entrepreneurship can contribute to the welfare of disabled people but can also contribute to society's welfare in general because it can generate net income for the State.

Analysing the ratios and variables such as age and number of employees, we see that these companies are socially sustainable. For the number of employees, it is evident that these companies contribute to social sustainability as they generate employment for people with disabilities. It should be recalled that at least 70% of staff in these companies must have a degree of disability equal to or greater than 33%. Thus, sheltered employment centres directly promote and generate social sustainability. Moreover, if we analyse the age of these companies, we can see that they have experience in the market. They have been operating for many years, and this creates more of a guarantee of employment for people with disabilities. At the same time, they can generate new jobs.

If we analyse the economic sustainability achieved by sheltered employment centres based on the descriptive analysis, we see that they also fulfil this function. The profitability

ratios show that they do not earn high profitability but achieve the minimum based on their activities. However, the ratio that measures the return to shareholders is quite high. In addition, the sales ratios show that these companies are economically sustainable in their core activities. Therefore, we can say that these companies actively contribute to social and economic sustainability [49,51].

*4.2. PART Algorithm*

To analyse the relationship between the selected independent variables and financial profitability (ROE), we developed a rule induction algorithm, specifically a PART algorithm. A PART algorithm [65] is a classifier that generates production rules by incorporating a modified form of the C4.5 decision tree and eliminating some of the paths found in an initial decision tree structure. Therefore, PART is a rule induction algorithm based on partial decision trees [66]. The main advantage of this classifier is its simplicity, as it develops the most general rule by choosing the leaf that represents the most significant number of situations. The rules obtained are logical sentences of the form 'if conditions then decisions', that is, the rules specify what decisions (actions) should be undertaken when some conditions are satisfied.

We have chosen ROE as the dependent variable because it measures the return on the capital provided by investors in terms of the net profit obtained in the year. It is an indicator of both economic and social return. The ROE variable has been codified as 1 for a positive return and 0 otherwise, to develop our model.

ROE is an important ratio for investors, allowing them to understand the creation of additional value [67], but also for entrepreneurs and managers because it helps them in the decision-making process [68–71]. With ROE, investors can assess if their investment is profitable or not, analysing the global efficiency rate [72]. We use ROE in this study because the government could be considered another "shareholder" when subsidies are paid in the same way as shareholders deposit their contributions.

To validate the rules, we have used a cross-validation procedure obtaining an 85.10% share of correctly classified sheltered employment centres (that is, accuracy). The satisfactory results in terms of accuracy allow us to interpret the strongest rules (see Table 6) to draw the following conclusions. First, there are more rules for class 1 than for class 0. This result indicates that it is easier to classify profitable firms (class 1) than unprofitable ones (class 0). There could be many different causes of unprofitability. Second, the key ratios to classify unprofitable companies would be liquidity and solvency (which would take negative or very low values) together with sales growth in this type of company. This means that this kind of firm is very conditioned by short-term ratios. Since they are mainly operating in the sector providing administrative activities and auxiliary services, they are dependent on cash. Third, the common ratios associated with solvent companies are the financial risk and indebtedness ratios. These ratios are related to the method of financing. They are more solvent if they are externally financed via banks. The reason is that banks only finance those businesses which are solid and have a market strategy and good leverage. A non-viable business does not receive funding from banks because financial institutions need guarantees about solvency and the repayment of the loans granted. Logically, if the companies have more debt, they have higher financial expenses. Thus, the two ratios (F-risk and Indb) are related. Sales per employee is also an important ratio for viable sheltered employment centres. As they are very labour intensive and have a social function to hire as many disabled workers as possible, the sales per employee ratio should be as high as possible (more than 11.5 in the case of the third rule, with 184 positive cases but 14 mistakes). Finally, according to the fourth rule, large service companies with less external financing are also profitable (178 positive cases against 41). These results are in line with the results of the descriptive analysis, which reinforces the argument about the economic sustainability of sheltered employment centres.

**Table 6.** PART rules.

|   | CURR-R | F-RISK | SOL | INDB | SALES-G | SALES-E | SECTOR | SIZE | DECISION | Strength (Corr/Incorr) |
|---|--------|--------|-----|------|---------|---------|--------|------|----------|------------------------|
| 1 | ≤0 |        | ≤0.1 |      | >−0.1 |        |        |      | 0 | (144/4) |
| 2 |   | >0 |     | >−2.4 |      |        |        |      | 1 | (205/2) |
| 3 |   | >−0.1 | ≤0.8 | >−2.4 | >−0.1 | >11.5 |        |      | 1 | (184/14) |
| 4 |   | >−0.1 |     | >−1 |      |        | 1 | >5 | 1 | (178/41) |

Source: own elaboration.

## 5. Conclusions

This paper aimed to test whether sheltered employment centres are sustainable over time from an economic and social perspective. To achieve our objective, we examined whether these firms could offset the public subsidies they receive for hiring people with disabilities through their profitability. We selected a sample of 997 Spanish sheltered employment centres, mainly micro-enterprises and small firms, with experience in the market and a wide range of employees, from 1 to more than 7000. Most of them are limited partnership companies located in extensive city areas, where most people live in Spain (Madrid, Andalusia and Catalonia). The main sectors in which these firms operate are administrative and auxiliary activities since these are the easiest to do in a routine and straightforward way. Between 2015 and 2019, the evolution of the main ratios for these firms led to an improvement in overall ROA.

In response to the first research question, we can say that sheltered employment centres do create and improve social and economic sustainability because they contribute more financially to the government than they receive. They are not highly profitable, but they survive in the market, hiring workers with diversity and giving them an economic and social role. In response to the second research question, we can say that sheltered employment centres do create social value through their operation and activity. If these firms receive external funding from banks, it means that they have a positive ROE and, consequently, are profitable. They therefore contribute to economic sustainability. The best companies are those operating in the service sector.

Our paper makes a clear contribution to examining the sustainability of social enterprises. These are profitable enterprises that can survive in the market, creating work for disadvantaged people. Instead of being at home and not having a role in society, workers with a disability can be in the workplace, feeling valuable. Moreover, these people being active could in turn reduce the cost of the health service as they are emotionally more engaged and could even be an example of effort and bravery for all workers. Their families could also feel that they are less dependent. Given our results, we encourage national, regional and local governments to invest in and financially support these firms, not only because they create jobs for disabled people but also because they are profitable.

This paper is not free from limitations. The sample is limited and only covers one country. Another limitation of the study is the lack of literature on social enterprises: each country has different types of social enterprises, and it is not possible to make comparisons between them taking into account their main objectives. The limited access to data about these firms is also a challenge, and future studies are needed to corroborate our conclusions. An enlargement of the sample and data on other countries is our task for future research. In addition, it would be interesting to use different methodologies to measure social value. Measuring social value by public data of company is difficult: almost all methodologies refer to internal data which in most cases is not possible to access. We believe that another better world is possible for people who have fewer opportunities, such as disabled workers.

**Author Contributions:** Conceptualization, V.G., M.J.S.-V., F.C.P.A. and M.d.M.C.-M.; methodology, V.G. and M.J.S.-V.; software, M.J.S.-V.; validation, V.G., M.J.S.-V., F.C.P.A. and M.d.M.C.-M.; formal analysis, M.J.S.-V.; investigation, F.C.P.A.; data curation, M.J.S.-V. and V.G.; writing—original draft

preparation, V.G; writing—review and editing, F.C.P.A.; supervision, M.d.M.C.-M.; project administration, M.d.M.C.-M.; funding acquisition, M.J.S.-V. and M.d.M.C.-M. All authors have read and agreed to the published version of the manuscript.

**Funding:** This research was supported in part by Universidad Complutense de Madrid under the Project Santander-UCM PR87/19–22586.

**Informed Consent Statement:** Not applicable.

**Data Availability Statement:** The data are available in the SABI database https://sabi.bvdinfo.com/version-2021531/home.serv?product=SabiNeo (accessed on 13 July 2021).

**Acknowledgments:** The authors would like to thank the work of all those volunteers and professionals who collaborate and work in the sheltered employment centers. Without them, people with functional or intellectual diversity would have much more difficult their professional performance. This work wants to serve as recognition, support and visibility to all of them.

**Conflicts of Interest:** The authors declare no conflict of interest.

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
