# Peer review of "Sheltered Employment Centres: Sustainability and Social Value"

_sustainability, doi:10.3390/su13147900_

Round 1

Reviewer 1 Report

This is an interesting study that does not ask for further adjustments regarding techical issues. But I would recommend to correct/review the English writing.

Author Response

We would like to thank the reviewer for reading the paper. Thank you for your positive feedback. As mentioned before, we have sent the manuscript to English editing services (as we have not the possibility to upload a separate document, we have put the proof of language editing at the end of this document).

Reviewer 2 Report

The literature review about the counter argument is not sufficient.

Further limitations of the research were well presented by the authors at the end of the paper.

Otherwise, the research methodology was appropriate with extensive use of available relevant data collected.

The subject chosen is original and sheds light to a contemporary social issue.

Author Response

Thank you for your comments. We agree the topic of our paper is original and can be a way to contribute to social business literature at the same time as a moment to make these kinds of firms visible in the economy. We have checked and enlarged the literature review section although there are not too many papers on this topic. We have added further limitations to our paper at the end of the manuscript.

“This paper is not exempt from limitations. The sample is limited and for only one country. Another limitation of the study is the lack of literature on social enterprises, each country has different types of social enterprises and it is not possible to make comparison between them taking into account their main objectives. The limited access to data of these firms is also a challenge, and more future studies are needed to corroborate our conclusions. The enlargement of the sample and data for other countries is our task for future research. Also, it would be interesting to use different methodologies to measure social value. Measuring social value by public data of company is difficult, almost all methodologies refer to internal data which in most cases is not possible to access. We believe that another better world is possible, overall for people with fewer opportunities as workers with disabilities are”.

….. “Taking that into account, we can say that the objective of social enterprises is to exhibit a greater sensitivity and promotion in the social and labor insertion of groups at risk of social exclusion compared to what ordinary companies should have [42]. In addition, social enterprises have a high impact at an international level, since they contribute to the social and labor insertion of the most needy people and prevent discrimination on the basis of gender [43]. It is also worth noting the importance of social enterprises in local and regional development as they can solve problems related to collective actions by creating networks and applying social norms to the situation [44]. Although, they are important companies, there are several barriers for these companies. The study elaborated by [45] identified the following barriers for growth of social firms: value based barriers (ethical value differences, growth philosophy, ethical principles), business models barriers (access to finance, access to human capital, identity authenticity), institutional barriers (consumer culture, business norms)”……….

…..“We choose the variable sales growth because research carried out by [64] has shown that social inclusion, as part of the Corporate Social Responsibility strategy, contributes in a positive way to sales growth”……

Reviewer 3 Report

Did the sheltered employment centres  have European financial support?

How did these results affect the analysed indicators?

The RQ2 hypothesis is not clearly stated in the analysis and conclusion phases.

The bibliography does not respect all the journal requirements.  

Author Response

Thank you for your questions. We have been investigating about the financial support of Sheltered Employment Centres and we found that the Sheltered Employment Centres receive subsidies through European funds. These subsidies are granted through projects that are managed by the autonomous regions, which are the entities that manage these resources in Spain. Consequently we have added it in the text in  introduction section 

“As sheltered employment centres receive many public aids from the government and other institutions such as the European Union”, ….. 

We have checked and rewritten the RQ2 hypothesis.

Do sheltered employment centres create social value through their operation and activity measured through the return to the public administrations?

Respect to the question about the question two, we discussed and it is not clear for us what about are you asking. We would be grateful if you can specify more what do you prefer and we will make the appropriate modifications.

Finally, we apologize for the bibliography formats. We have modified it in order to adjust all the journal requirements. 

Reviewer 4 Report

Thank you very much for the invitation to review the article. The problem which shown this article is interesting and important.     I think that the content of Table 3. is not very clear.    The part which includes the discussion should show the implication in a broad context. There are no references to the previous research.

Author Response

Thank you for your positive feedback about our paper and for the suggestions.

Related to the content of Table 3, we agree. It is not clear enough. We have added the word “location” to the first column of data and “industry” in the next column. We have explained in depth this table. The idea of Table 3 is to summarize the location and the industry of the sheltered employment center of our sample. We have explained better. We have clarified the content of Table 3 adding a paragraph before the table:

….”In Spain there are 17 autonomous regions and there is a different regulation about grants and subsidies for the CEEs for each autonomous region. Therefore, the variable location is important because it can condition the subsidies received by the CEEs”……

We have added references to prior research in the discussion section in order to enrich this section. Thank you for the comments.
